# DPP-IV Inhibitory Peptides GPF, IGL, and GGGW Obtained from Chicken Blood Hydrolysates

**DOI:** 10.3390/ijms232214140

**Published:** 2022-11-16

**Authors:** Gisela Carrera-Alvarado, Fidel Toldrá, Leticia Mora

**Affiliations:** Instituto de Agroquímica y Tecnología de Alimentos (CSIC), Avenue Agustín Escardino 7, 46980 Paterna, Spain

**Keywords:** DPP-IV inhibitor, hypoglycemic peptides, type 2 diabetes mellitus, enzymatic hydrolysis

## Abstract

Blood is a meat by-product rich in proteins with properties that can be improved after hydrolysis, making it a sustainable alternative for use in the generation of bioactive peptides. The objective of this study was to identify dipeptidyl peptidase IV (DPP-IV) inhibitory peptides obtained from different chicken blood hydrolysates prepared using combinations of four different enzymes. Best results were observed for AP (2% Alcalase + 5% Protana Prime) and APP (2% Alcalase + 5% Protana Prime + 3% Protana UBoost) hydrolysates obtaining inhibition values of 60.55 and 53.61%, respectively, assayed at a concentration of 10 mg/mL. Free amino acids were determined to establish the impact of exopeptidase activity in the samples. A total of 79 and 12 sequences of peptides were identified by liquid chromatography and mass spectrometry in tandem (LC-MS/MS) in AP and APP samples, respectively. Nine of the identified peptides were established as potential DPP-IV inhibitory using in silico approaches and later synthesized for confirmation. Thus, peptides GPF, IGL, and GGGW showed good DPP-IV inhibitory activity with IC_50_ values of 0.94, 2.22, and 2.73 mM, respectively. This study confirmed the potential of peptides obtained from chicken blood hydrolysates to be used as DPP-IV inhibitors and, therefore, in the control or modulation of type 2 diabetes.

## 1. Introduction

Type 2 diabetes mellitus is a common metabolic disorder, with a prevalence of 10.5% of adults worldwide [1]. It is characterized by elevated blood glucose levels due to complete or relative insufficiency of insulin secretion [2]. After food intake, glucagon-like peptide 1 (GLP-1) and gastric inhibitory polypeptide (GIP) secrete about two-thirds of total insulin [3]. However, both peptides are natural substrates of the enzyme DPP-IV and are degraded rapidly in vivo (1–2 min), causing the loss of their insulinotropic activity. Therefore, inhibition of the DPP-IV enzyme is an important strategy for the effective treatment of diabetes [4]. DPP-IV inhibitors aim to prolong the half-life of endogenous GLP-1 and GIP and, consequently, increase their plasma concentrations to decrease the postprandial blood glucose level [5,6]. Due to their biological origin, hypoglycemic peptides are considered safer as a treatment for diabetes, as they have fewer side effects than synthetic drugs [7,8]. Currently, the two most important agencies that regulate the access of drugs to the market, the Food and Drug Administration of the United States (FDA) and the European Medicines Agency (EMA) have approved more than 90 peptides as treatments for various diseases [9]. An example of these is lixisenatide (Lyxumia^®^), a polypeptide of 44 amino acids residues approved in 2013 by the EMA indicated for the treatment of diabetes.

Food bioactive peptides are generated from protein hydrolysis that occurs during the processing, cooking, and/or digestion of food. However, they can also be obtained through controlled hydrolysis with commercial proteolytic enzymes and according to their length and amino acid sequence, some of such peptides released may be bioactive [10]. There is evidence that some food protein hydrolysates inhibit DPP-IV in silico, in vitro, and in vivo. For instance, α-lactalbumin hydrolysate inhibits DPP-IV activity with an IC_50_ value of 0.036 mg/mL [11]. A study with a hydrolyzed protein isolate (HPI) of black beans and their pure peptides reported that glucose absorption decreased 21.5% after 24 h of treatment (10 mg HPI/mL) in a Caco-2 cell model, and in male Wistar rats a 24.5% reduction in postprandial glucose (50 mg HPI/kg body weight) was found [12]. A simulated duodenal digestion study with quinoa protein and fraction > 5 kDa, obtained after 120 min of digestion showed high inhibition of DPP-IV and its IC_50_ was 0.84 mg protein/mL [13].

The development of bioactive peptides from inexpensive and readily available protein raw materials is currently being promoted [14], like by-products from the meat industry such as blood, bones, viscera, skin, organs, etc., with the aim of adding value to protein-rich raw materials and improving the sustainability of meat production. For example, blood represents approximately 3 to 5% of body weight [15], in addition to its nutritional value, it contains proteins with technological functions or bioactive substances that can be generated by enzymatic hydrolysis [16]. Hydrolysates derived from dietary proteins are generally 10^4^ times less potent than gliptins, however, the combination of gliptins and hydrolysates may enhance the effect on their ability to inhibit DPP-IV [17].

Therefore, the objective of this work was to determine those DPP-IV inhibitory peptides obtained from the most active chicken blood hydrolysates using chromatographic separations and the subsequent identification of peptide sequences by mass spectrometry in tandem. The identified peptides were studied using in silico approaches and those sequences showing the best potential as DPP-IV inhibitors were evaluated to confirm their bioactivity.

## 2. Results and Discussion

### 2.1. Degree of Hydrolysis and Inhibitory Activity of DPP-IV

Enzymatic protein hydrolysis is extensively used in the food industry to enhance the functional properties and nutritional quality of products, such as the elimination of allergenicity, flavor production, detoxification, etc. [18,19]. The degree of hydrolysis of chicken blood hydrolysates is shown in Figure 1. The *DH* increased up to three times more when applying sequentially two or more enzymes and a longer hydrolysis time (16 h), being AFP (2% Alcalase + 1% Flavourzyme + 3% Protana UBoost) and AP (2% Alcalase + 5% Protana Prime), the samples with the highest *DH*, 30.29 and 29.51%, respectively. It is considered that in a process of sequential hydrolysis, performing a pre-digestion with alcalase increases the number of N-terminal sites available for the subsequent action of some exo-peptidases as it occurs in our study with Flavourzyme or Protana Prime [19].

Glycemic regulation and energy homeostasis can be improved by maintaining the incretin effect by inhibiting DPP-IV [20]. At a concentration of 10 mg/mL, the DPP-IV inhibition rate of the hydrolysates AP and APP showed the highest results (*p* < 0.05), with values of 60.55 and 53.61%, respectively. In the case of A hydrolysate, despite having the lowest degree of hydrolysis, it presented a DPP-IV inhibition rate of 39.38%. Regarding the hydrolysates AF and AFP, they showed DPP-IV inhibition values of 38.56 and 44.50%, respectively (Figure 1). Multiple sequential hydrolyses can produce peptides with improved or reduced activities [21]. In addition, it is known that Flavourzyme achieves aggressive hydrolysis, that could result in the hydrolysis of the bioactive peptide sequences obtained and decrease their bioactivity [7]. Bioactive peptides are released through different kinetics; the larger ones appear in the initial phase of hydrolysis and are cut into smaller peptides that, according to their characteristics, can show different bioactivity [22]. In addition, it has been reported that the DPP-IV inhibitory activity of a hydrolysate is determined by its peptide structure and sequence, which depends largely on the type and action of proteases used during hydrolysis [23]. For example, in a study with sheep serum hydrolysates, the inhibitory activity of DPP-IV was independent of the degree of hydrolysis, and they attributed the higher rate of inhibition (77.80%) to the characteristics of the peptides produced by the enzyme trypsin [8]. Mojica & de Mejía [24] reported that the best conditions to generate antidiabetic peptides from black bean protein isolate was by hydrolysis with Alcalase for 2 h and 1:20 enzyme/substrate, with a DPP-IV inhibition rate of 96.7%. In our study, AP and APP were hydrolyzed with Alcalase and Protana Prime, the first is an endo-protease, and the second is an exo-peptidase, two specific enzymes for the generation of peptides and small peptides/free amino acids, respectively. The inhibitory activity of DPP-IV has been attributed mainly to small peptides. According to the BIOPEP-UWM database, near 59% of the registered DPP-IV inhibitory peptides are dipeptides (Figure 2A).

Due to AP and APP hydrolysates showing the best inhibitory results, these samples were selected for the subsequent separation into peptide fractions using RP-HPLC, and the DPP-IV inhibitory activity of each fraction was determined. From the HPLC chromatogram, the elution was separated into 20 main fractions according to the elution time (0–40 min). Of the 20 fractions tested of both hydrolysates, fraction 2 (elution time 3–4 min), corresponding to polar peptides, exhibited the highest inhibitory activity of DPP-IV with values of 44.01% in AP and 37.33% in APP (Figure 3). While in the rest of the chromatogram non-significant values of DPP-IV inhibition were observed. Consequently, 54.40% of the 432 DPP-IV inhibitory peptides registered in BIOPEP-UWM are hydrophilic (<0) and the rest are hydrophobic (<0.50) (Figure 2B). Therefore, these fractions were collected again, lyophilized, and analyzed by LC-MS/MS to identify the sequences of DPP-IV inhibitory peptides contained in fraction 2 of the AP and APP hydrolysates.

### 2.2. Determination of Free Amino Acids (FAAs)

The protein content in food constitutes the source of essential amino acids in the diet. It has been reported that the bioactivity of antidiabetic peptides depends especially on their amino acid profile and their function is mainly based on inhibiting the activity of the DPP-IV enzyme [4]. Figure 4 shows the free amino acid content of hydrolysates A, AP, and APP. As expected for an endo-protease, the hydrolysis with Alcalase produced a low content of free amino acids; however, when combined with Protana Prime, a significant amount of amino acids were generated. Lys, Leu, Ala, Val, and Glu were found in greater concentration. A significant decrease in Gln was also found in AP with respect to APP hydrolysates, whereas, APP had a significantly higher Glu content than AP. These results are attributed to the specific hydrolytic action of Protana UBoost, which is based on the conversion of glutamine into glutamic acid. Xu et al. [25] state that a first hydrolysis with Alcalase allows several peptide bonds to be accessible and especially recognizes the Ala, Leu, and Val sites without affecting the residues of Pro amino acids in the hydrolysate, which has probably occurred in our hydrolysates. Similar to our results, the content of free amino acids in porcine blood hydrolysates increases after the enzymatic hydrolysis process, including branched-chain amino acids such as Val, Ile, and Leu [26,27]. Sadri et al. [28] affirm that Leu and Val are involved in glycemic control, and Pro may increase the hypoglycemic activity of peptides.

### 2.3. In Silico Analysis of the Identified Peptides

The amino acid sequences of the peptides contained in fraction 2 of the AP and APP samples were identified by LC-MS/MS. A total of 79 and 12 peptide sequences were identified in AP and APP fractions, respectively. These peptides were studied using the Peptide Ranker tool to establish those sequences more able to exert DPP-IV inhibitory activity, obtaining 17 and 1 in AP and APP, respectively, with values greater than 0.5 (Appendix A). After simulating gastrointestinal digestion using in silico tools, novel peptides were generated, of which 21 have been previously described as bioactive with inhibitory activities of DPP-IV (18 peptides), ACE-I (13 peptides), renin (3 peptides), and antioxidant (3 peptides). Regarding DPP-IV inhibitors, peptides such as AT, GPA, GPAG, VN, AW, and PM are located at the N- or C-terminal of the identified AP hydrolysate sequences, which could also be active in their initial form before the simulated digestion (Table 1).

According to previous reports, some of the characteristics of DPP-IV inhibitory peptides are (i) hydrophobicity in nature, (ii) length from 2 to 8 amino acids, and (iii) Pro residue located between the first four N-terminal positions [22,29,30], flanked by Gly, Phe, Leu, Val, and Ala [31]. In addition, Power et al. [6] mentioned that the DPP-IV enzyme also cleaves, although to a lesser degree, peptides containing Ser, Gly, Val, and Leu in the N-terminal position. Therefore, according to their amino acid sequence and physicochemical characteristics, a total of nine peptides generated in the simulated gastrointestinal digestion were selected as potential DPP-IV inhibitors and later synthesized: Gly-Trp (GW), Ala-Trp (AW), Gly-Pro-Phe (GPF), Gly-Gly-Gly-Trp (GGGW), Ile-Gly-Leu (IGL), Pro-Met (PM), Pro-Leu (PL), Ile-Phe (IF), and Cys-Phe (CF). The best results are shown in Figure 5, where only those peptides with significant positive results were represented. In particular, three peptides showed outstanding inhibitory activity, with values of 65.19% (GPF), 43.64% (IGL), and 40.74% (GGGW) at a concentration of 2 mM. These results confirm the previously reported information as peptides GGGW, GPF, and GW contain Gly as an N-terminal residue, although GPF peptide contains a Pro residue in the second N-terminal position and is also flanked by Gly and Phe, which makes it the best potential DPP-IV inhibitor. Regarding hydrophobicity values shown in Table 2, they ranged from 0.21 to 0.47, since sequences were mainly composed of hydrophobic amino acids such as Trp, Ala, Ile, and Leu, making them potential sources of hypoglycemic peptides. According to the DPP-IV inhibition rate values, the IC_50_ of the three peptides with the highest inhibition was determined in vitro. The values obtained were 0.94 mM for GPF, 2.22 mM for IGL, and 2.73 mM for GGGW (Table 2, Figure 5). In the same trend, in a study with salmon skin gelatin hydrolysates, two peptides GPAE and GPGA were identified as DPP-IV inhibitory peptides, with IC_50_ values of 49.6 and 41.9 µM, respectively [23]. LPYPY, a peptide identified in milk proteins from in silico digestion with gastrointestinal proteins, was also identified as a potent DPP-IV inhibitor, with an IC_50_ value of 108 µM [30]. These peptides contained Pro as the second N-terminal residue, and the Pro residue was flanked by Ala, Gly, or Leu. In addition, the peptides were mainly composed of hydrophobic amino acid residues, such as Ala, Gly, and Pro. However, these peptides show a lower IC_50_ value than those obtained in this study, probably due to the primary sources from which they were obtained.

## 3. Materials and Methods

### 3.1. Chemicals and Reagents

The Alcalase, Flavourzyme, Protana UBoost, and Protana Prime were purchased from Novozymes (Bagsvaerd, Denmark). Ile-Pro-Ile, o-phthaldialdehyde (OPA), Gly-Pro-7-amido-4-methylcoumarin (Gly-Pro-AMC) bromhydrate, trifluoroacetic acid (TFA), and DL-dithiotreitol (DTT) were from Sigma-Aldrich (Saint Louis, MO, USA). Dipeptidyl peptidase IV (from porcine kidney, EC 3.4.14.5), sodium dodecyl sulfate (SDS), and 2-mercaptoethanol were from Merck (Darmstadt, Germany). The synthetic peptides Gly-Pro-Phe (GPF), Gly-Gly-Gly-Trp (GGGW), Ile-Gly-Leu (IGL), Gly-Trp (GW), Ala-Trp (AW), Pro-Met (PM), Pro-Leu (PL), Ile-Phe (IF), and Cys-Phe (CF) were obtained from Bachem. Tris-(hydroxymethyl) aminomethane and formic acid (FA) were from Panreac Química S.A. (Barcelona, Spain). The supergrade HPLC grade acetonitrile (ACN) and supergrading HPLC grade methanol (MeOH) were from Scharlau Chemie (Barcelona, Spain). All other chemicals and reagents used were of analytical grade.

### 3.2. Enzymatic Hydrolysis

The hydrolysis process was carried out sequentially. In the first step, Alcalase, an endo-protease with an aggressive hydrolytic action, was used. In the second step, Flavourzyme and Protana Prime were used. Flavourzyme exerts moderate endo- and exo-peptidase activity being responsible for generating peptides and amino acids, whereas Protana Prime is mainly an exo-peptidase enzyme capable to generate free amino acids. In a third step, Protana UBoost was used with its main activity as a glutaminase enzyme. Thus, 1 g of boiled chicken blood was diluted in 2 mL of bidistilled water (500 mg/mL), brought to 80 °C for 15 min and a sequential hydrolysis process was applied. The first hydrolysis was prepared with 2% Alcalase (A) for 2 h at 55 °C. Subsequently, hydrolysate A was carried out to the second hydrolysis with 1% Flavourzyme (AF), 1% Flavourzyme + 3% Protana UBoost (AFP), 5% Protana Prime (AP), or 5% Protana Prime + Protana UBoost (APP) for 16 h at 55 °C. The digestion was stopped using heat in boiling water for 10 min. Finally, the hydrolysates were stored at −20 °C. All samples were prepared in triplicate.

### 3.3. Degree of Hydrolysis Determination

The OPA solution was made by mixing the following reagents: 50 mL of 100 mM sodium tetraborate decahydrate, 5 mL of 20% SDS (*w*/*w*), 80 mg of OPA dissolved in 2 mL of MeOH, 200 μL of 2-mercaptoethanol and adjusted to a final volume of 100 mL with bidistilled water. Each hydrolysate was diluted a hundred times in bidistilled water. For the reaction, 100 μL of the sample was incubated with 3.4 mL of OPA reagent for 2 min at room temperature. Absorbance was measured at 340 nm (Cary 60 UV-visible spectrophotometer, Agilent Technologies, CA, USA). The degree of hydrolysis (*DH*) was calculated according to the following equation:$$DH\;\left(\%\right)=\frac{\left(ABS*1.934*d\right)}{c}$$
where *ABS* is the absorbance of the samples, *d* is the dilution factor and *c* is the protein concentration of the sample (g/L) [32]. All measurements were prepared in triplicate.

### 3.4. DPP-IV Inhibitory Activity

The DPP-IV inhibition assay was performed following the methodology proposed by Gallego et al. [5]. Briefly, in 96-well microplates, 45 μL of inhibitor, control, or sample (10 mg/mL) and 45 μL of DPP-IV at 5 mU/mL (diluted with 50 mM Tris-HCl + 5 mM of CaCl2 + 1μM of ZnCl2, pH 8.0) and 180 μL of 0.25 mM Gly-Pro-AMC (diluted with 50 mM Tris-HCl buffer + 0.5 mM DTT, pH 8.0). Fluorescence generation was measured in a CLARIOstar multimode microplate reader (BMG LABTECH, Ortenberg, Germany) using excitation wavelengths of 355 nm and emission of 460 nm at 0 and 20 min with an incubation temperature of 37 °C. Ile-Pro-Ile was used as a reference inhibitor and 50 mM Tris-HCl buffer (pH = 8) was used as a control. The percentage of DPP-IV inhibition was calculated using the following formula:$$\%\;Inhibition=\frac{Control\;fluorescence\;\left(t_{20}-t_{0}\right)-sample\;fluorescence\left(t_{20}-t_{0}\right)}{Control\;fluorescence\;\left(t_{20}-t_{0}\right)}\times 100$$

### 3.5. Peptide Separation by RP-HPLC

The hydrophobicity of the peptides was studied by reverse phase HPLC. A Symmetry C18 column (4.6 × 250 mm, 5 μm, Waters Co., Milford, MA, USA) was used. As solvent A 0.1% TFA in bidistilled water was used and as solvent B, 0.085% TFA in ACN (ACN:H_2_O, 60:40, *v*/*v*). A 100 μL sample was injected at a concentration of 100 mg/mL. The elution was controlled at 214 nm. The peptides were run in a step gradient mode, where solvent B remained constant for 2 min, then it reached 50% in the next 50 min. The flow rate was 1 mL/min. Fractions were collected every two minutes. Each fraction was lyophilized and then re-dissolved in 50 μL of bidistilled H_2_O to determine its DPP-IV inhibitory activity as indicated in Section 3.4.

### 3.6. Determination of Free Amino Acids (FAAs)

For sample preparation, 300 μL of hydrolysates A, AP, and APP were homogenized with HCl 0.01 N (1:4; *w*/*v*) in a vortex for 8 min and centrifuged at 10,000 g and 4 °C for 20 min. For the analysis of FAAs, the samples were pretreated following the methodology described by Aristoy & Toldrá [33], which includes chemical deproteinization and derivatization of the sample. A 5 mM norleucine solution was used as the internal standard. The chromatographic separation of the derivatized molecules was done following the methodology described by Flores et al. [34]. Briefly, using a reverse phase HPLC system (Series 1200; Agilent, Santa Clara, CA, USA), equipped with a Waters Pico Tag^®^ C18 column (3.9 × 300 mm; Waters Corp., Milford, MA, USA) at a temperature of 52 °C and flow rate of 1 mL/min. Elution was monitored at 254 nm. As phase A, 70 mM sodium acetate with 2.5% ACN, pH 6.55 was used. As phase B, ACN:H_2_O:MeOH was used in a ratio of 45:40:15. Quantification was carried out considering the response factors calculated for each amino acid/sample in the mixed standards series. The results were expressed as mg of FAAs/100 mL of hydrolysate.

### 3.7. Peptide Sequence Identification by LC-MS/MS

A 1 µL sample was injected into an Ekspert nanoLC 425 (Eksigent, Redwood City, CA, USA) using a trap column C18-CL (3 µ 120 Ᾰ, 350 m × 0.5 mm; Eksigent), where it was desalted and concentrated with a mobile phase of 0.1% TFA at a flow of 5 µL/min for 5 min. Subsequently, the peptides were loaded onto a C18-CL analytical column (3 µ, 120 Ᾰ, 0.075 × 150 mm; Eksigent) equilibrated with 5% ACN and 0.1% FA. Solvent A was 0.1% FA dissolved in bidistilled water, and solvent B was ACN in 0.1% FA. Elution was carried out using a linear gradient of 15 to 40% of B in A for 20 min at a flow rate of 300 nL/min. Peptides were then analyzed in a mass spectrometer nanoESI qQTOF (6600 plus TripleTOF, ABSCIEX), where the peptides were ionized in a Nano Optiflow Source Type applying 3.0 kV to the spray emitter at 200 °C. Mass spectrometry was carried out in the data-dependent acquisition mode. MS1 scans of 350–1400 *m*/*z* were obtained for 250 ms. The quadrupole resolution was set to “LOW” for MS2 experiments, which were acquired from 100–1500 *m*/*z* for 25 ms in high sensitivity mode. The criteria used for switch was a charge from 2+ to 5+, minimum intensity, and 100 cps. Up to 100 ions were selected for fragmentation after each survey scan. Dynamic exclusion was set to 15 s. Digestion of 500 ng K562 trypsin was used as a control for system sensitivity. Regarding the data analysis, the obtained spectra were analyzed using ProteinPilot v 5.0. (SCIEX) default parameters. The peak list was directly generated from 6600 plus TripleTOF wiff files. The Paragon algorithm of ProteinPilot v 5.0 was used to search the Uniprot_GallusGallus database with no enzyme specificity.

### 3.8. In Silico Analysis of Identified Peptides

Peptide Ranker tool was used to predict the potential bioactivity of the peptides [35], and values greater than 0.5 were considered for additional in silico analyses simulating gastrointestinal digestion. Gastrointestinal digestion was simulated using the BIOPEP-UWM database [36], using the enzymes chymotrypsin (EC 3.4.21.1), trypsin (EC 3.4.21.4) and pepsin (EC 3.4.23.1) accessed on 12.05.22. The potential allergenicity of the synthesized peptides was predicted using the AllerTOP v. 2.0 software [37]. ToxinPred tool [38] was accessed on 7 July 2022 to study the toxicity of peptides and their physicochemical properties (hydrophobicity, amphipathicity, pI, and steric hindrance).

### 3.9. Statistical Analysis

For data analysis, an analysis of variance (ANOVA) followed by Tukey’s study range test was used, with a significance level of *p* < 0.05. Each data point represents the average of three samples.

## 4. Conclusions

The hydrolysates AP (2% Alcalase + 5% Protana Prime) and APP (2% Alcalase + 5% Protana Prime + 3% Protana UBoost) have a significant DPP-IV inhibitory activity (60.55% and 53.61%, respectively, assayed at a concentration of 10 mg/mL). The peptides GPF, IGL, and GGGW were identified after simulated GI digestion of the peptides identified in the fraction with the highest DPP-IV inhibitory activity, showing IC_50_ values of 0.94, 2.22, and 2.73 mM, respectively. These results confirm the potential of chicken blood hydrolysates as a source of bioactive peptides as GPF, IGL, and GGGW could be good candidates to be used in the control or modulation of type 2 diabetes, although in vivo analysis would be needed in order to confirm their antidiabetic activity.

## Figures and Tables

**Figure 1 ijms-23-14140-f001:**
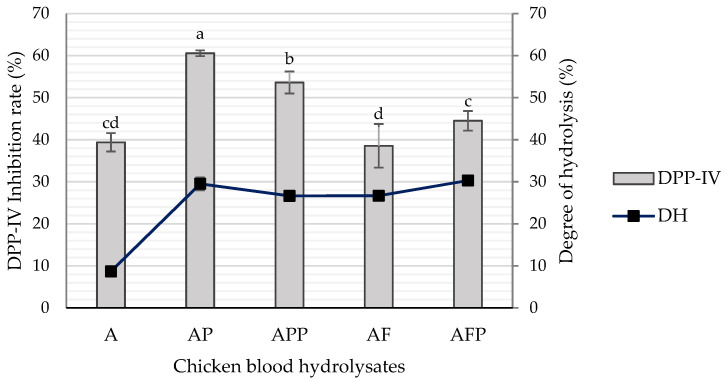
DPP-IV inhibition rate and degree of hydrolysis of different chicken blood hydrolysates: A (2% Alcalase), AP (2% Alcalase + 5% Protana Prime), APP (2% Alcalase + 5% Protana Prime + 3% Protana UBoost), AF (2% Alcalase + 1% Flavourzyme) and AFP (2% Alcalase + 1% Flavourzyme + 3% Protana UBoost). Bars with the same letter between samples are not statistically different. Tukey, *p* < 0.05.

**Figure 2 ijms-23-14140-f002:**
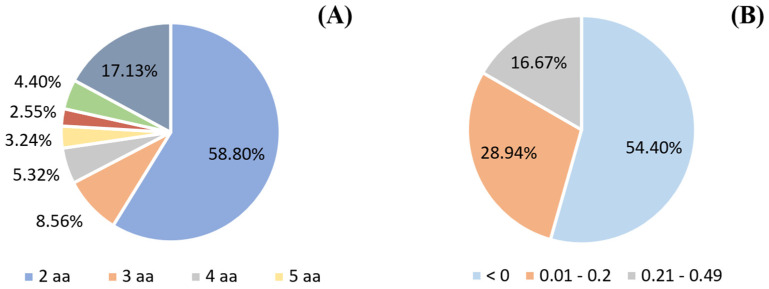
(**A**) DPP-IV inhibitory peptides contained in the bioactive peptides database BIOPEP-UWM grouped according to their amino acid (aa) length. (**B**) DPP-IV inhibitory peptides grouped according to their hydrophobicity, where low values indicate high hydrophobicity and higher values indicate low hydrophobicity. Data obtained from ToxinPred tool.

**Figure 3 ijms-23-14140-f003:**
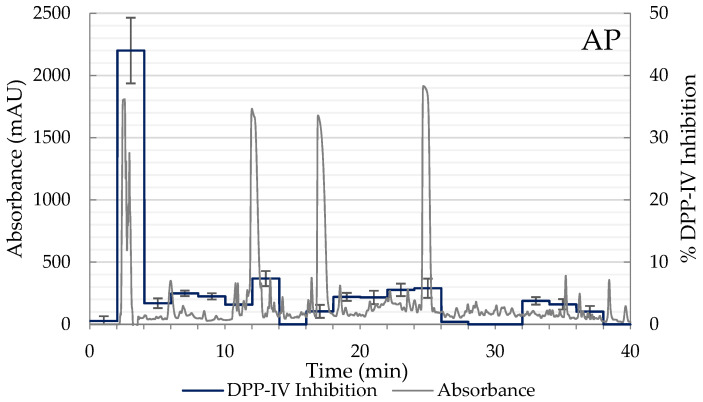
Elution profile and DPP-IV inhibition rate of the peptide fractions of the hydrolysates AP (2% Alcalase + 5% Protana Prime) and APP (2% Alcalase + 5% Protana Prime + 3% Protana UBoost) separated by RP-HPLC. The inhibition rate of DPP-IV was determined with each fraction at a concentration of 100 mg/mL hydrolysate.

**Figure 4 ijms-23-14140-f004:**
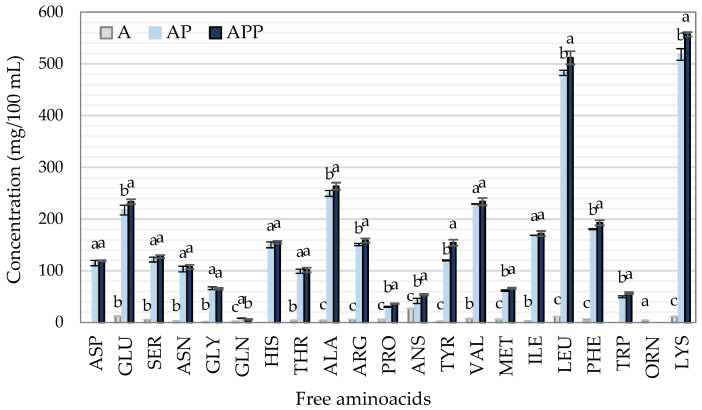
Concentration of free amino acids in three different hydrolysates: A (2% Alcalase), AP (2% Alcalase + 5% Protana Prime) and APP (2% Alcalase + 5% Protana Prime + 3% Protana UBoost). Bars with the same letter between samples are not statistically different. Tukey, *p* < 0.05.

**Figure 5 ijms-23-14140-f005:**
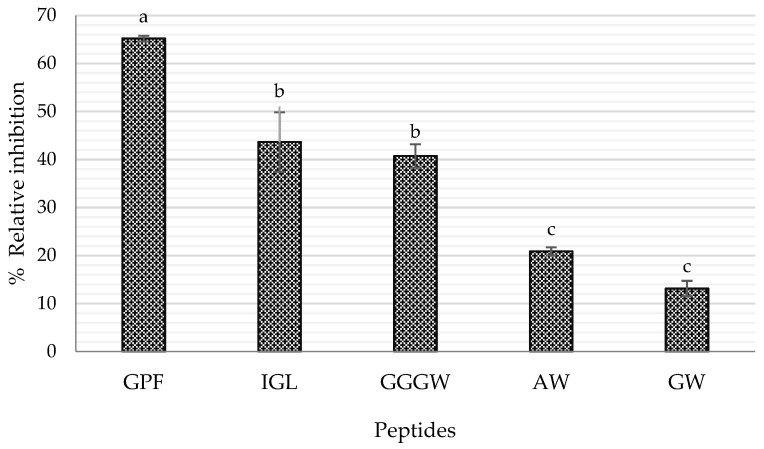
Relative DPP-IV inhibition rate of peptides at 2 mM. Bars with the same letter between samples are not statistically different. Tukey, *p* < 0.05.

**Table 1 ijms-23-14140-t001:** In silico study of the identified peptides in chicken blood hydrolysates.

Sample		Peptide Ranker ^1^	Peptide Sequence	MW (g/mol)	Enzyme Action ^2^	Active Fragments	Bioactivity
**AP**		0.971435	GNGGGWGNSGGGGGGGGGGNGDGGGGDGCGNSGGCGGGGG	2926.42	GN–GGGW–GN–SGGGGGGGGGGN–GDGGGGDGCGN–SGGCGGGGG	-	-
**AP**		0.892122	NCFAAGF	728.90	N–CF–AAGF	CF	ACE inhibitor
**AP**		0.864842	PTWALLGCVLLLPSLR	1752.44	PTW–AL–L–GCVL–L–L–PSL–R	AL	Dipeptidyl peptidase IV inhibitor
**AP**		0.861194	FIGLFIGISKFMAT	1545.13	F–IGL–IGISK–F–M- AT	AT	Dipeptidyl peptidase IV inhibitor
**AP**		0.833521	LGPFSFSFGPAT	1227.54	L–GPF–SF–SF–GPAT	SF	ACE inhibitor, dipeptidyl peptidase IV inhibitor and renin inhibitor
**AP**		0.715609	PVGPMGPLGPA	992.36	PVGPM–GPL–GPA	GPL–GPA	ACE inhibitor and dipeptidyl peptidase IV inhibitor
**AP**		0.704105	GPAGDAGAEGKPGIPG	1350.68	GPAGDAGAEGK–PGIPG	-	-
**AP**		0.690027	AWIRYSKVKFVSFNF	1892.68	AW–IR–Y–SK–VK–F–VSF–N–F	AW	ACE inhibitor, antioxidative, dipeptidyl peptidase IV inhibitor
						IR	ACE inhibitor, antioxidative, renin inhibitor, CaMPDE inhibitor and dipeptidyl peptidase IV inhibitor
						SK	Dipeptidyl peptidase IV inhibitor
						VK	ACE inhibitor, dipeptidyl peptidase IV inhibitor
**AP**		0.671755	GAGLLLLEALEKGYWV	1732.31	GAGL–L–L–L–EAL–EK–GY–W–V	EK–GY	ACE inhibitor, dipeptidyl peptidase IV inhibitor
**AP**		0.670104	GEKGPLGPNGPVGV	1277.66	GEK–GPL–GPN–GPVGV	GPL	ACE inhibitor
**AP**		0.63634	SPSKRFRGWRARTERT	1991.45	SPSK–R–F–R–GW–R–AR–TER–T	GW	ACE inhibitor, dipeptidyl peptidase IV inhibitor
						AR	ACE inhibitor
**AP**		0.607254	RDGPQGPLGPAG	1121.39	R–DGPQGPL–GPAG	GPAG	Dipeptidyl peptidase IV inhibitor
**AP**		0.571682	GPQGKVGPTGAPG	1122.44	GPQGK–VGPTGAP	-	-
**AP**		0.560064	GPAGAPGFPGAPGSKGEAGPTGARG	2121.65	GPAGAPGF–PGAPGSK–GEAGPTGAR -G	-	-
**AP**		0.551038	RAAELRPLR	1081.40	R–AAEL–R–PL–R	PL	ACE inhibitor and dipeptidyl peptidase IV inhibitor
**AP**		0.518296	PMADSGCLTEGEMGLIFVN	1984.57	PM–ADSGCL–TEGEM–GL–IF–VN	PM	Dipeptidyl peptidase IV inhibitor
						GL	ACE inhibitor and dipeptidyl peptidase IV inhibitor
						IF	ACE inhibitor
						VN	Dipeptidyl peptidase IV inhibitor
**AP**		0.50915	FSFLPQPPQEKAHDGGRYY	2237.73	F–SF–L–PQPPQEK–AH–DGGR–Y–Y	SF	ACE inhibitor, dipeptidyl peptidase IV inhibitor and renin inhibitor
						AH	ACE inhibitor, antioxidative and dipeptidyl peptidase IV inhibitor
**18**	APP	0.584356	KCYTPVCLK	1054.45	K–CY–TPVCL–K	-	-

^1^ To predict the bioactivity potential of peptides, the Peptide Ranker tool was used to predict the potential bioactivity of peptides. ^2^ Gastrointestinal digestion was simulated with the BIOPEP-UWM database using the enzymes chymotrypsin (EC 3.4.21.1), trypsin (EC 3.4.21.4), and pepsin (EC 3.4.23.1).

**Table 2 ijms-23-14140-t002:** Main physicochemical characteristics attributed to the peptides in this study.

Characteristics	GPF	GGGW	IGL	AW	GW
MW (g/mol)	319.39	375.44	301.43	275.32	261.30
Charge	0.00	0.00	0.00	0.00	0.00
Isoelectric point (pI)	5.88	5.88	5.88	5.88	5.88
Steric hindrance	0.58	0.64	0.64	0.51	0.59
Sidebulk	0.58	0.64	0.64	0.51	0.59
Hydrophobicity	0.23	0.21	0.47	0.31	0.27
Hydrophilicity	−0.83	−0.85	−1.20	−1.95	−1.70
Amphipathicity	0.00	0.00	0.00	0.00	0.00
IC_50_ (mM)	0.94	2.73	2.22		

Physicochemical property values obtained from ToxinPred.

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
