# Peer review of "DPP-IV Inhibitory Peptides GPF, IGL, and GGGW Obtained from Chicken Blood Hydrolysates"

_ijms, 2022, doi:10.3390/ijms232214140_

Round 1

Reviewer 1 Report

This manuscript investigated the use of chicken blood for the production of DPP-IV inhibitory peptides. The study was well designed. Minor revision is suggested for further improve the quality of the manuscript.

How about these values comparing to what have been reported in literature?

Line 128-130: the authors should specify that those with minus values (<0) indicating hydrophilicity in Figure 2B?

Figure 3: suggest to change the bar graph to another format. The shadow was somehow distracting.

Table 1: what are the parent sources of these peptides?

Line 181-199: The DPP-IV inhibitory activities need comparisons with other reported peptides.

Measurement of these peptides at the concentration of 2 mM showed relatively mild inhibition of DPP-IV, thus the IC50 values of 0.94 – 2.73 mM, which was hard to reach in vivo. This needs further comments.

Reviewer 2 Report

This paper described the study of DPP IV inhibitory peptides from chicken blood hydrolysates.

Authors used synthetic peptides to determine the IC50 value. The biggest question of this paper is which blood proteins did the peptide sequences come from? The authors need to clarify this point.

I think that the peptide ranker shows the bioactivity of an antimicrobial peptide, which is different from DPP-IV.

What is the most abundant protein in blood?

What is the protein from which the peptide is derived?

Please identify the origin of the peptide sequence by blast or other software.

Although the method 4.7. indicated MS range was 100-1500 m/z, many peptides in the table 1have lengths outside this range. How did the authors detect the peptides?

Please add the mass of peptide in table 1.

Round 2

Reviewer 2 Report

In the introduction mentions the use of by-products for peptide sources. The authors focused on the minor protein components. Therefore, I do not fully understand the significance of this study since hemoglobin is the most abundant protein in blood. It seems to me that a large amount of by-products arise from by-products.